# Accuracy of a Bayesian technique to estimate position and activity of orphan gamma-ray sources by mobile gamma spectrometry: Influence of imprecisions in positioning systems and computational approximations

**Antanas Bukartas**[1]*, **Jonas Wallin**[2], **Robert Finck**[1], **Christopher Rääf**[1]

**1** Medical Radiation Physics, Lund University, Lund, Sweden, **2** Department of Statistics, Lund University, Lund, Sweden

* antanas.bukartas@med.lu.se

## Abstract

The purpose of this study was to investigate the effects of experimental data on performance of a developed Bayesian algorithm tailored for orphan source search, estimating which parameters affect the accuracy of the algorithm. The algorithm can estimate the position and activity of a gamma-ray point source from experimental mobile gamma spectrometry data. Bayesian estimates were made for source position and activity using mobile gamma spectrometry data obtained from one 123% HPGe detector and two 4-l NaI(Tl) detectors, considering angular variations in counting efficiency for each detector. The data were obtained while driving at 50 km/h speed past the sources using 1 s acquisition interval in the detectors. It was found that deviations in the recorded coordinates of the measurements can potentially increase the uncertainty in the position of the source 2 to 3 times and slightly decrease the activity estimations by about 7%. Due to the various sources of uncertainty affecting the experimental data, the maximum predicted relative deviations of the activity and position of the source remained about 30% regardless of the signal-to-noise ratio of the data. It was also found for the used vehicle speed of 50 km/h and 1 s acquisition time, that if the distance to the source is greater than the distance travelled by the detector during the acquisition time, it is possible to use point approximations of the count-rate function in the Bayesian likelihood with minimal deviations from the integrated estimates of the count-rate function. This approximation reduces the computational demands of the algorithm increasing the potential for applying this method in real-time orphan source search missions.

## Introduction

Lost ionizing radiation sources (orphan sources) must be rapidly brought back under regulatory control to avoid the risk of harm to the public and the environment [1–3]. It may be necessary to search for lost gamma-ray sources over large areas and long distances. Vehicle-borne

**Data Availability Statement:** https://github.com/SpontaneousFusion/AUTOMORC_Data.

**Funding:** This work was financially supported by the Swedish Radiation Safety Authority (SSM 2016/2274) and by the Nordic Nuclear Safety Research. Grant received by professor Christopher Rääf. The funders had no role in study design, data collection and analysis, decision to publish, or preparation of the manuscript.

**Competing interests:** The authors have declared that no competing interests exist.

mobile gamma spectrometry is often used to make measurements in a sequence of short time intervals (1–10 seconds) along public roads [4]. A gamma-ray source can be detected by the increase in the observed count-rate as the vehicle passes the source. This provides an indication that a radioactive source is nearby, but gives no direct information on the actual position or activity of the source. However, this information can be extracted from the measurement time-series collected while driving past the source along the road. One way to achieve this is to use Bayesian analysis.

There has been many studies regarding the applicability of Bayesian or similar statistical inference methods to mobile gamma spectrometric data regarding source localisation—e.g. utilising additive point-source localisation algorithms for sparse parametric image reconstruction [5, 6], reconstructing position of the source using arrays or networks of stationary radiation detectors positioned in an area [7–10], utilising Bayesian Aggregation to combine multiple measurement data and learning the expected signal-to-noise ratio trend [11], or using coded apertures to suppress the influence of background increasing the maximum distance for detecting sources [12].

In our previous study [13] we have performed a feasibility test of a Bayesian algorithm using simulated data from multiple detectors with individual angular variations in counting efficiency of the detectors, estimating position and activity of point gamma emitting sources, with distances to the sources spanning the range of 10-190 m and activities reaching 1215 MBq.

Factors that significantly influence the precision with which a gamma-emitting source can be located using mobile gamma spectrometry system within the aforementioned distances between the source and the detector and activities, include attenuation of the gamma particles in the line of sight between the detector system and the source, fluctuating natural background count rate (as addressed by e.g. [14–17]), and gain drift of the radiation detectors. Furthermore, in emergency situations, the resources available for orphan source searches may be limited in terms of time, manpower and equipment [18]. In addition, the most commonly used search algorithms are based on simple statistical analysis of count rates in selected spectral windows in the gamma spectrum [19, 20], but these require additional measurements, from backpack- or handheld instruments to more accurately localize the source (e.g. [21]). This can be a drawback if the search is to be conducted in areas with restricted physical access, and it may be very difficult or sometimes impossible to determine the position of the orphan source in a limited amount of time.

From the previous study [13] it was concluded that using data from multiple detectors provided more information for the algorithm and, as a result, the estimates of the position and the activity were more accurate. This study was purely theoretical, and it is thus important to investigate how well the algorithm performs using real data, which is associated with additional uncertainties (such as varying background count rates) and technical limitations (e.g. inconsistent Global Navigation Satellite System (GNSS) signal, delays in communication between the detector and computer, gain drift of the detectors, etc.). The aim of this study is to investigate the performance of the Bayesian algorithm using actual field measurement data from mobile gamma spectrometers. It is conducted in terms of estimates of the locations and activities of gamma-ray point sources located at different distances from the road with varying activities, while determining which parameters influence the accuracy of these estimates.

## Theory

Bayesian inference is a statistical method using Bayes theorem for combining prior knowledge with new information, resulting in posterior probability distributions of the parameters of

interest. When searching for an orphan source, the parameters of interest are the position and the activity of the source so, in this case, the result of Bayesian inference would be the posterior probability distributions of the position and the activity of the source. To be able to perform Bayesian inference, a Bayesian model must be constructed, consisting of two constituents: a prior and a likelihood. The information that is available before performing the measurements and the Bayesian inference constitute the prior, describing the knowledge about the parameters of interest by assigning them selected probability distributions. The likelihood, on the other hand, describes how well the data is supported by the chosen model and selected values of parameters of interest.

When searching for an orphan source the information available regarding the position and activity of the source may be incomplete. In such situations a "vague" prior is usually used, consisting of probability distributions with large variance, allowing the likelihood to dominate the posterior distribution. If there is reason to believe that a radioactive source is in the area, it can be assumed that it could be anywhere in the area, thus the prior for the position of the source in two dimensions could be set up using uniform probability distributions for $x$ and $y$ coordinates of the source. This denotes equal probability of the source being at any position $x$ and $y$. Similarly, a vague Gamma distribution can be set up as a prior for activity, allow to pass some information regarding the activity of the source to the algorithm while still allowing the likelihood to dominate the posterior. The Gamma distribution was set up with shape and rate parameters set to 1.2 and 0.0001 correspondingly, while the activity was scaled down by a factor of $1e6$, basically allowing any positive values of activity.

In the context of searching for an orphan source, the likelihood describes the probability of the data collected according to the selected Bayesian model describing the physics of gamma-ray interaction in matter, and selected values of parameters of interest. This allows gamma-ray detection knowledge to be utilized in the Bayesian inference, which is usually not included in traditional methods of searching for orphan sources. It is well known that the count-rate function of a radiation detector, $\dot{N}(r)$, can be described by the activity of the gamma source, $A$, the branching ratio, $n_\gamma$ denoting the number of gamma photons emitted per decay, the distance between the source and the detector, $r$, the counting efficiency of the detector, $\varepsilon$, the coefficient of linear attenuation of gamma photons in air, $\mu_{\text{air}}$ and the background count rate $c$:

$$\dot{N}(r) = \frac{A n_\gamma \varepsilon \exp(-\mu_{\text{air}} r)}{4\pi r^2} + c. \tag{1}$$

Although, in reality the background count rate $c$ can vary significantly throughout a particular area, the background count rate within this study is regarded as a constant due to relatively stable background radiation within the area where the experiment was performed.

Providing the radioactive source is of sufficient activity and close enough to the trajectory of the detector, there might be a visible peak in the measurement time series of the detector after the detector has passed a radioactive source. The width and height of this peak depends on the distance and activity of the source. By performing Bayesian inference on the shape of the peak it is possible to obtain estimations of the position and activity of the source.

Gamma detectors may have different geometric shape and have an inert material in the vicinity, for example photomultiplier tube, electronic components and detector casing that will affect the efficiency of the detector. If the detector is mounted in a vehicle, structural components of the vehicle can provide variable amounts of additional shielding. Because of this, the counting efficiency of a detector mounted in a vehicle can be quite different for different angles of incidence of gamma photons $\theta$. Thus, the angular variations of the counting efficiency were taken into account in the likelihood to make the calculations more accurate. As

demonstrated in [13], it is possible to express the relative angle of incidence $\theta$ using the current $x_i$, previous $x_{i-1}$ measurement coordinates and the position of the source $p$. Thus, for an $i$-th measurement, a simplified final equation depicting the physical model for the likelihood yields:

$$\dot{N}_i(x, p, A) = \frac{An_\gamma \varepsilon(x_i, x_{i-1}, p)\exp(-\mu_{\mathrm{air}}\|x_i - p\|_2)}{4\pi\|x_i - p\|_2^2} + c. \tag{2}$$

where $x$ is the measurement position and $p$ is the source position in two spatial coordinates. Additionally, individual shift of each detector from the position of the GNSS receiver was also taken in to account.

In the context of searching for an orphan source, the prior becomes a probability distribution for the position of the source, $P$, and the activity of the source, $A$,—written $\pi(P, A)$. Similarly, the likelihood can be expressed as a probability distribution of measurement values, $Z$, provided that the measurement locations, $X$, position, $P$, and activity, $A$, of the source are known, $\pi(Z|X, P, A)$. The likelihood was adapted to use data from multiple detectors in the Bayesian calculations simultaneously to increase the accuracy of the results, as was displayed in a previous study [13]. The likelihood $\pi(Z|X, P, A)$ can then be then expressed as:

$$\pi(Z|X, P, A) = \prod_{i=1}^{n}\prod_{j=1}^{m}\mathrm{Pois}(z_{i,j}|\dot{N}_{i,j}(x, p, A)) \tag{3}$$

where $m$ is the number of detectors and $n$ is the number of measurements.

It is then possible to combine the measurements (observations) with knowledge (likelihood and priors) and using the Bayes formula evaluate the posterior distribution up to a normalizing constant:

$$\pi(P, A|Z, X) \propto \pi(Z|X, P, A) \cdot \pi(P, A), \tag{4}$$

where $\pi(P, A|Z, X)$ is the posterior probability distribution of position $P$ and activity $A$ of the source, given measurement values $Z$ obtained at measurement locations $X$. In essence, Bayesian inference utilizes the likelihood to produce probability values of the parameters of interest, according to the selected model and input data, and then "filters" the resulting probability distribution using the prior.

As the target posterior distribution is a complex distribution, it cannot be directly inferred. Thus, to obtain the samples representing the target posterior distribution indirect sampling was performed using Markov Chain Monte Carlo (MCMC) integration, written in language and environment for statistical computing R [22, 23]. Functions within default package "stats" version 3.6.3 was used to evaluate distributions and draw random numbers.

To obtain a sample from the target distribution, a set of initial guess values of position and activity of the source are chosen first. Then, for each MCMC iteration the following process is repeated: generation of a proposition of the parameters of interest, calculation of the acceptance ratio for the new proposition compared to the previous value, and then comparison of the acceptance ratio to a generated uniform random number ($u \in [0; 1]$). If the acceptance ratio is higher than the random number $u$, the proposition is accepted. Otherwise—previous sample is kept. After a number of iterations the obtained distribution of samples will be proportional to the target distribution. As it takes time for the chain to converge to the target distribution, the first samples obtained might be representative of a different distribution. Thus, a *burn-in* of the MCMC chain is typically used—a number of the first iterations of the chain is discarded.

For a more detailed description of the Bayesian model, the reader is referred to a previous study [13].

## Materials and methods

The experiment was performed on a relatively straight stretch of one-lane road (approximately 1.7 km long) near the former Barsebäck nuclear power plant, about 20 km north of the city of Malmö, in southern Sweden. The data were collected using a mobile gamma spectrometry system in a vehicle equipped with ionizing radiation detectors (a HPGe detector with 123% relative efficiency and two 4-l NaI(Tl) detectors), multichannel analyzers, computer and a GNSS unit (Fig 1).

The HPGe detector was positioned in the centre of the back portion of the service-bed section of the vehicle. Two 4-l NaI(Tl) detectors were mounted one after the other under the roof on the right side of the service-bed compartment of the vehicle (referred to as the front and rear NaI(Tl) detector). The experiment involved 20 different experimental set-ups, in which point sources of $^{137}$Cs and $^{60}$Co were used, with five separate activities for each radionuclide.

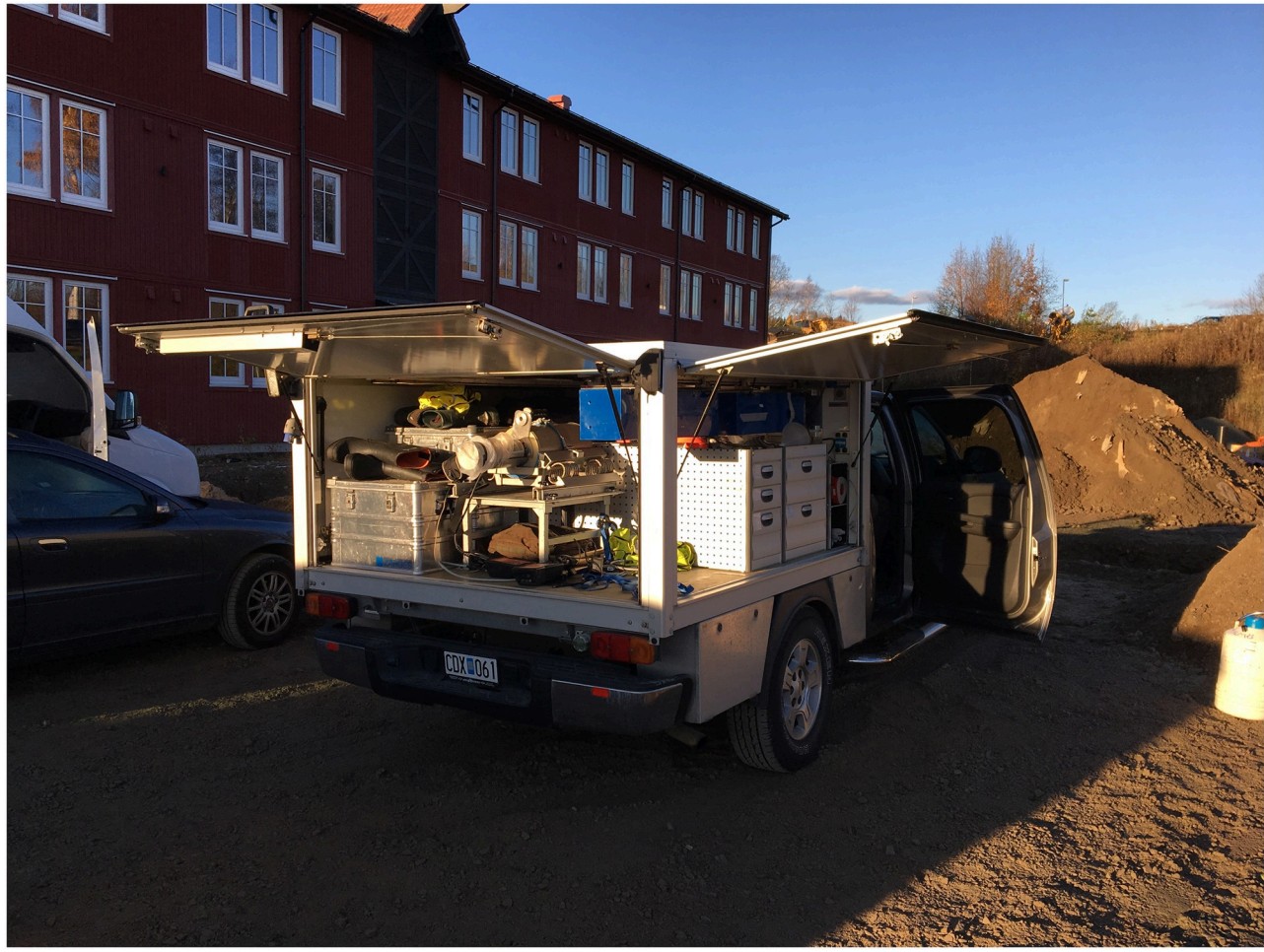

**Fig 1. A picture of service-bed compartment of the gamma spectrometry vehicle used in the experiment.** The 123% relative efficiency HPGe detector with its nitrogen dewar is visible in the centre-back of the service-bed. The two blue boxes in the upper-right part of the service-bed are the two 4l NaI(Tl) detectors. The GNSS receiver was mounted on the roof of the cabin, above the front passenger.

## Overview of the vehicle path

**Fig 2. Illustration of the path of the vehicle and the positions of the radioactive sources.** The symbol × marks all of the positions of the $^{137}$Cs and $^{60}$Co sources used in the experiment. The latitude and the longitude are geographic coordinates reported in accordance to World Geodetic System standard WGS 84.

The experimental set-ups were divided into groups of four. In each group the sources had the same activity, but were gradually moved further away from the roadside. The vehicle was then driven along the road past the two radioactive sources, placed at selected distances from the road side. A schematic view of the path of the vehicle and all of the positions of the sources used throughout the experimental set-ups is shown in the Fig 2.

The exact distances from the roadside and the activities of the sources in all the experimental set-ups are shown in Fig 3. The actual distances between the sources and the detectors while the vehicle was passing the sources were about two metres longer, because the distances to the sources were measured to the road side and that the vehicle had to be driven in the appropriate lane of the road while passing the source. Exact distance to the source depends on the lane of the road the vehicle was driving—the source was closer to the detectors while passing the sources on the right side than the left side of the vehicle.

The sources were placed in the defined positions for about 10 minutes for each experimental set-up. During this time, the gamma spectrometry vehicle was driven back and forth along the road, passing the radioactive sources several times. One complete pass is defined as driving the vehicle from one turning point to the other, past the radioactive sources, at a steady speed of 50 km/h (13.9 m/s). At least 7 complete passes of the sources were made for each experimental set-up. To keep the analysis consistent, only the data from the first seven passes of the sources were used in further analysis. The Bayesian estimations of position and activity of the source were performed for each complete pass separately to investigate the potential of the Bayesian algorithm in time-limited situations. The individual complete pass of an experimental set-up will be referred to as a segment further in the text.

Gamma spectrometric data were collected each second during the passage of the segment. Regions of interest (ROIs) in the collected spectra were selected covering the primary gamma peaks in the spectra of the $^{137}$Cs source (661.6 keV line) and the $^{60}$Co source (1332.5 keV line). The number of counts in the selected ROIs were integrated for every gamma spectrometric acquisition. For each new spectrometric acquisition the geographical position of the

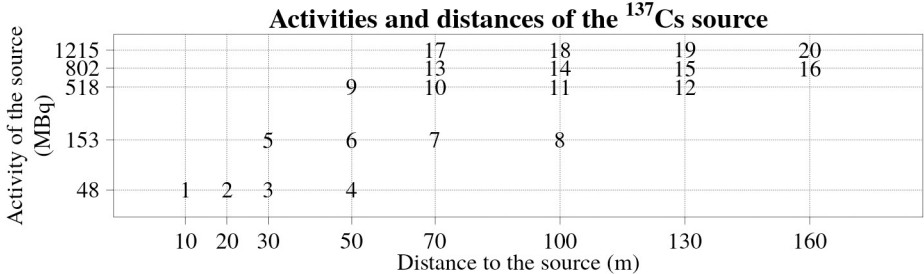

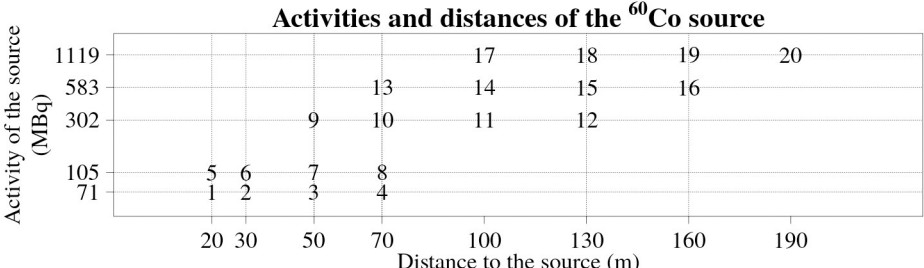

**Fig 3. Distances from the roadside to the $^{137}$Cs and $^{60}$Co sources and activities of the sources used in the experiment for every experimental set-up.** The numbers in the graphs represents the experimental set-up number. The actual distance from the sources to the detectors were a couple of metres longer.

measurement was recorded using the GNSS system. The measurement positions and gamma spectrometric data were recorded using Nugget software, developed by the Swedish Radiation and Safety Authority [24]. These integrated ROI values with the corresponding geographical coordinates were then used in the Bayesian algorithm.

During the analysis of the data, a slow gain drift was observed in the 4-l NaI(Tl) detectors. As the amount of relative gain drift during a particular experimental set-up was negligible, drift compensation was performed using a single scaling value to realign the energy of the $^{40}$K gamma peak (1460.8 keV line) to the $^{40}$K peak in the spectrum time series for all of the data of a particular experimental set-up. For a detailed description of the calibration of counting efficiency of the detectors with angular variations, the reader is referred to a previous study [13].

### Estimation of the position and activity of a point gamma source using the Bayesian algorithm

The position and activity of the sources for each individual segment were estimated using the Bayesian algorithm developed in previous study [13]. Three datasets were analysed: data from the HPGe detector only (henceforth denoted HPGe), data from the rear NaI(Tl) detector only (NaI) and data from all three of the detectors (the HPGe detector and the two 4-l NaI(Tl) detectors)(Multi). The Bayesian algorithm was run for 30 000 MCMC iterations. The first 10 000 iterations were discarded as burn-in. The points with the highest probability in posterior probability distributions of source position and activity were considered to be the estimated positions and activities.

As the angle of incidence of the gamma photons was not recorded in the mobile gamma spectrometry system, the algorithm could not determine which side of the road the radiation originated from, thus resulting in probable positions of the source on both sides of the road, i.e., a bimodal posterior distribution of position of the source. To determine the discrepancy in distance between the estimated position and the actual position of the source correctly in such

situation, the actual position of the source was mirrored onto the other side of the road, and the distances between the estimated position and the actual position, and that between the estimated position and the mirrored position were calculated. The smaller of the two values was then used in the analysis.

The estimated positions and activities for individual segments were analysed in groups according to the radionuclide, detector combination and experimental set-up. Relative deviations of the estimated positions and activities were calculated for each of the seven segments in each experimental set-up. The relative deviations in the position of the source were additionally separated into a lateral (perpendicular to the road) and a longitudinal (along the road) deviations. The relative deviations $RD$ were then calculated as below, where EST is the estimated value, and the REF reference value:

$$\mathrm{RD} = \frac{(\mathrm{EST} - \mathrm{REF})}{\mathrm{REF}}. \tag{5}$$

In the Eq 5 above, the REF reference values for position, lateral and longitudinal relative deviations were distances between the source and the roadside for corresponding radionuclides in the experimental set-ups. Respectively, the reference values for activity relative deviations were actual activity of the source. The median values of the relative deviations were then calculated throughout each experimental set-up, detector and source combination. These median values were then used in further analysis.

## Assessment of point approximation of the count-rate function in the detector

Usually, during orphan source search missions the spectrometry system is configured to output integrated count-rate during a selected acquisition time interval $t_{\mathrm{acq}}$. Using Eq 1, it is possible to express the average number of counts in the detector during this time interval $t_{\mathrm{acq}}$:

$$N(t) = \int_{t}^{t+t_{\mathrm{acq}}} \frac{A n_{\gamma} \varepsilon \exp(-\mu_{\mathrm{air}} r(t))}{4\pi(r(t))^2} + c. \tag{6}$$

When a detector is approaching a radioactive source in a straight trajectory, the count-rate function will increase due to decreasing source-detector distance $r$. When the minimum source-detector distance is reached, the count-rate function obtains it's maximum value. When detector is continuing to travel past the radioactive source, the distance will start to increase resulting in a decrease in the count-rate function. This increase and then decrease in the count-rate while a detector is passing a source gives rise to a peak within the count-rate function. An example is visible in Fig 4 for a situation, when a detector passes a radioactive source positioned 10 metres from the trajectory of the detector using $t_{\mathrm{acq}}$ = 1 s travelling at a speed of 50 km/h.

The count-rate function and the acquisition time intervals can be aligned in various ways depending on the distance to the source, length of the acquisition time interval $t_{\mathrm{acq}}$, speed of the vehicle and position at which the acquisition procedure has started. The actual alignment can be anything in between the best and worst alignment situations, displayed in Fig 5. These, best and worst alignment situations are the most extreme cases of the relative alignments of the count-rate function and the acquisition time intervals, resulting in the most pronounced difference between the registered number of counts after passing a source for both relative alignment situations.

To correctly estimate the number of counts during the acquisition interval thus the count-rate function needs to be integrated during the selected interval length $t_{\mathrm{acq}}$. Although it is

**Integrated and point estimate comparison**

**Vehicle speed 50 km/h (13.9 m/s), distance to source 10 m**

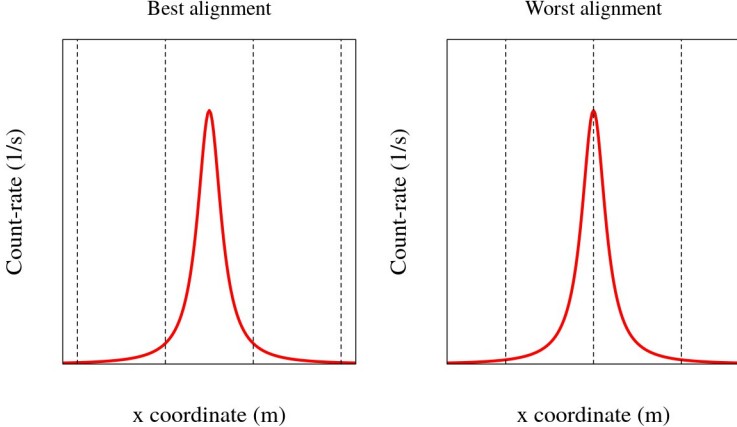

**Fig 4. Comparison of point vs integrated estimates of measurement values of mobile gamma spectrometry data when the source is positioned at 10 m.** The red line denotes the countrate function in the detector, black circles mark the point estimates and green squares—integrated estimates. Relative difference of the areas under the point and integrated estimate peaks is 2.79%.

simple to evaluate a finite sum integral of a count-rate function using a computer, it takes significantly longer than the calculation of a point estimate of that function (depending on the number of summing iterations). When using the MCMC method to sample from the posterior distribution, integrals of the count-rate function would have to be calculated for each measurement during each MCMC iteration, significantly increasing the amount of time and computational resources needed to obtain the posterior distribution. If point estimates of the count-rate function performed in the middle of the acquisition interval could be used to estimate the number of counts in the detector, this additional computational burden could be avoided.

Comparison of integrated and point estimates of the count-rate function for best alignment situation is displayed in Fig 4. Calculating the relative difference of areas under the peaks in

**Relative alignment of the acquisition interval and the count-rate function**

Best alignment · Worst alignment

**Fig 5. Visualisation of relative alignment of the count-rate function and the acquisition time intervals.** On the left a best alignment situation is displayed, when the the detector was the closest to the source during the acquisition interval. On the right, worst alignment situation is displayed, when the closest distances to the source are divided between two adjacent acquisition time intervals.

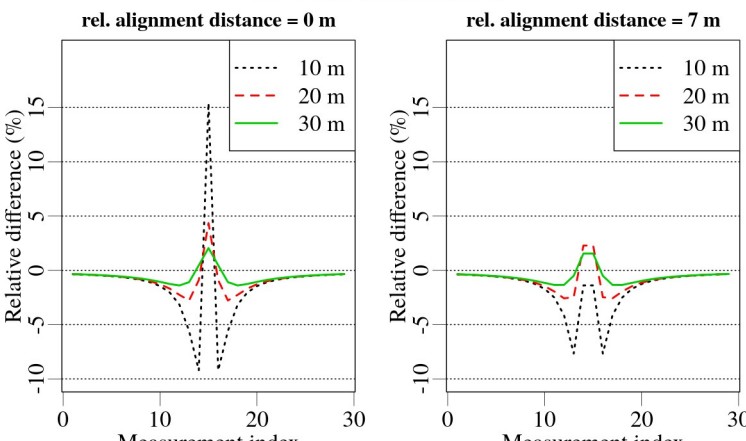

**Fig 6. Comparison of relative point-by-point differences between integrated and point estimates of the count rate function for different offsets and distances to the source.**

measurement time-series displayed in Fig 4, yields a small difference of 2.79%. Point-by-point differences of the count-rate function estimates are considerably larger, reaching 15%. These point-by-point differences are shown in Fig 6 for two different relative alignments of the measurement acquisition intervals and the count-rate function for source distances of 10, 20 and 30 m. As can be seen in Fig 6, the different relative alignments has a significant effect on the point-by-point differences; the greatest deviations being seen for the closest (10 m) sources in the best alignment situation. Already at a distance of 20 m, the maximum absolute point-by-point difference for the best alignment situation has decreased by about 3 times, from 15% to less than 5%, while the reduction for the worst alignment was smaller—from 7% to 3%. The discrepancy between the point-by-point differences for varying relative alignments arises from the distance difference between the source and the detector at the start and the end of the acquisition time interval. Because more time during the acquisition time interval in the best alignment situation is spent closer to the source than in the worst alignment situation, a more significant difference in the point-by-point variation occurs.

Bearing in mind that the differences between the point and integrated estimates are significant only for distances of 10 m and less when using an acquisition time of 1 s while driving at a speed of 50 km/h, it was decided to continue to use point estimates in the Bayesian model as it is hoped that the method will ultimately be applicable in real-time use. This could affect the estimates in experimental set-up 1, where a $^{137}$Cs source was placed at a distance of 10 m.

Thus, calculations were performed to compare point and integration estimates of the count-rate function in the Bayesian model for experimental set-up 1. The integrated estimates in the likelihood calculations use 11 iteration finite sum integration. Posterior distributions for the comparison were obtained after 30 000 MCMC iterations with a burn-in of 10 000 iterations. Results of the comparison are displayed in the Results section.

## Effects of the discrepancy in the coordinates of the measurement position on the Bayesian estimates

Measurement coordinates are usually obtained using a GNSS system such as GPS, Galileo, GLONASS, etc., which can be provided in various formats at selected time intervals. In the

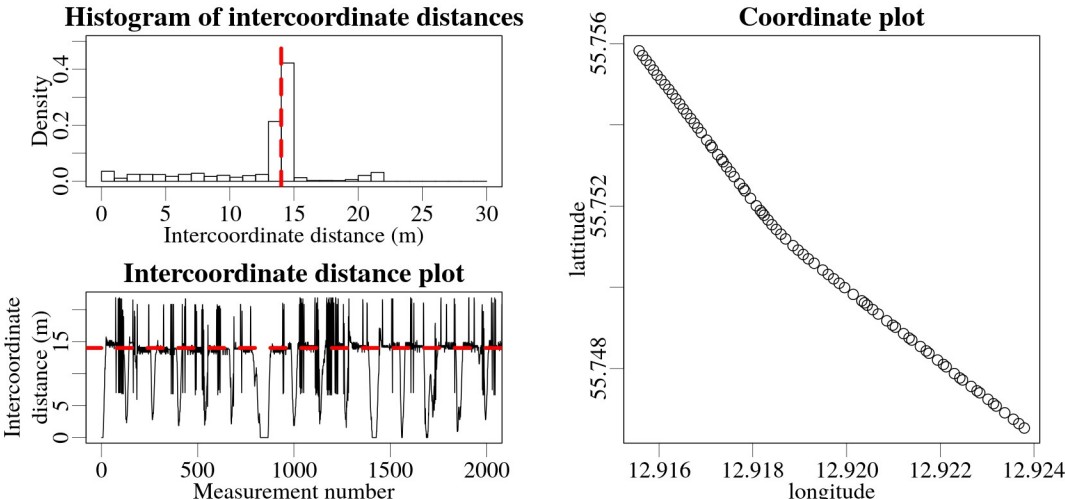

**Fig 7. Intercoordinate distance analysis of the GNSS coordinates recorded and processed by the Nugget software for experimental set-up 1.** The upper graph shows a histogram of the length of intercoordinate distances. The lower graph shows the intercoordinate distance values for each measurement. The average speed of the vehicle during the experiment was 50 km/h (13.9 m/s), indicated by the red dashed lines.

mobile gamma spectrometry system used in this study, the Nugget software reads the data from the navigation system and the detectors, and then saves and analyses the data and visualises the results for each new measurement. Due to peculiarities in how Nugget is programmed, the timing of the read-out of the coordinates from the navigation system is affected, which manifests as discrepancies between the recorded and the actual measurement coordinates. This can be observed in the analysis of the distances between adjacent measurement coordinates (intercoordinate distances) for experimental set-up 1 in Fig 7. During the experiment, the vehicle was driven back and forth along the road past the sources at the speed of about 50 km/h (13.9 m/s). At the end of a complete pass, the vehicle was then turned around and another pass of the sources was made.

The discrepancies can be seen as deviations in the measurement coordinate along the trajectory of the vehicle (longitudinally), equivalent to adding or subtracting half of the intercoordinate distance to the actual measurement position along the path of the vehicle, implying that the speed of the vehicle was fluctuating by about 7 m/s. If the discrepancies were to occur in the vicinity of a radioactive source during a passage of the source, the recorded count rates could be assigned incorrect geographical coordinates, resulting in a distorted shape of the peak in the measurement time-series. This could potentially affect the estimated position and activity of the source.

To roughly evaluate the potential effects of such discrepancies on the Bayesian estimations, the estimations were performed for generated synthetic data with and without simulated discrepancies in the geographical coordinates. First, a simplified mechanism of coordinate deviations was reverse engineered from the experimental data. The data obtained while speeding up and slowing down of the vehicle (increases and decreases in the intercoordinate distances visible in Fig 7) were filtered out, keeping only the data recorded while travelling at a constant speed of 50 km/h. It was observed that in many cases the coordinate discrepancies in the experimental data occurred in small groups, affecting several successive recorded positions. As an approximation, it was chosen to simulate the synthetic discrepancies as groups of six individual consecutive deviations. The probability of the group of deviations occurring with the

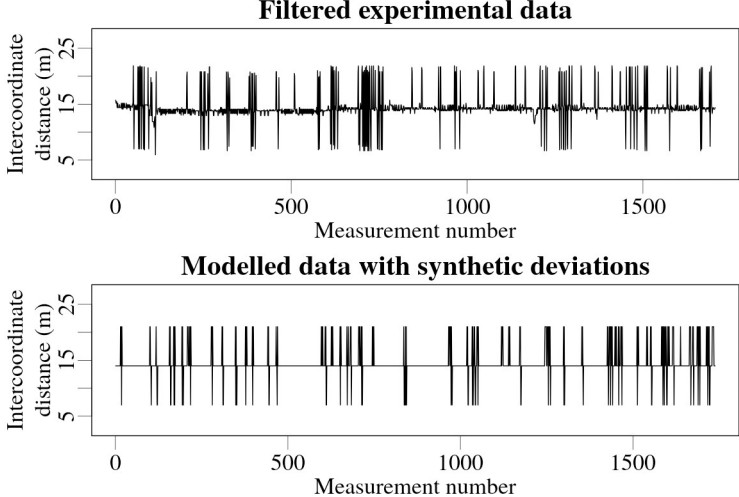

**Fig 8. Filtered experimental data from experimental set-up 1 with visible deviations (top), and modeled data with simulated deviations, based on the probabilities obtained from the data above (bottom).**

first deviation increasing the intercoordinate distance, was calculated. Probabilities for consecutive individual deviations within the group being larger, the same as, or lower than the actual intercoordinate distance were also evaluated from the experimental data. Using these calculated probabilities the synthetic geographical coordinates were then altered to approximate the kind of discrepancies observed in the experimental data. Intercoordinate distances obtained in experimental data and then roughly replicated in the modelled data for one of the experimental set-ups are shown in Fig 8.

A number of synthetic measurement coordinate points were established along a 2 km long straight path, with distance between the measurement points set to 13.9 m corresponding to the 50 km/h speed of the vehicle used in the experiment. Synthetic measurement data due to a 100 kBq $^{137}$Cs radioactive source positioned at a 30 m distance from the trajectory of the detector was calculated using the physical model described in the Theory section. To eliminate the effect of statistical variations in the data, average count-rate values instead of samples from a Poisson distribution were used. These average values represent infinitely long measurements with eliminated statistical uncertainty.

Using the method to introduce the discrepancies within measurement coordinates discussed above, 20 different realizations of the coordinate discrepancies for the synthetic coordinate data were calculated. Bayesian estimates were then evaluated for the synthetic data with the discrepancies within the coordinates using combined posterior distributions from 20 different MCMC chains—running 30 000 iterations each, meaning a total of 600 000 iterations. First 10 000 iterations of every chain were discarded as a burn-in. Then, Bayesian estimates were evaluated for the synthetic data without the coordinate discrepancies, again using 20 different MCMC chains. The posterior distributions for source position and activity obtained while using the discrepancies were then compared with the posterior distributions obtained using synthetic data without the discrepancies. Results of the comparison are displayed below.

It has to be noted, that this is only a rough approximation of the discrepancy mechanics, visualising the potential effects on the Bayesian estimations, which can be expected to occur due to similar discrepancies within the recorded measurement coordinates.

## Evaluation of the influence of the signal-to-noise ratio of the data on the relative deviations of the estimated position and activity

Signal-to-noise ratios (SNRs) of the data were calculated for every combination of detector, source and experimental set-up. The SNRs were calculated using a formula:

$$\mathrm{SNR} = \frac{I - B}{\sqrt{B}}. \tag{7}$$

where $I$ is a single value of maximum average number of counts in a selected ROI of gamma spectrum, covering the full energy peak of primary gamma photons of interest, within measurement time-series obtained during a passage of radioactive source, and $B$ is the background level within that ROI. The values obtained are shown in Fig 9.

The median values of the relative deviations for each source, experimental set-up and detector combination were then plotted against their respective SNRs. When combinations of detectors were considered, the total combined SNR was obtained as a linear combination of the values for the respective detectors. It is expected that there will be an inverse relation between the SNRs and the magnitude of the relative deviations—i.e. a higher SNR of the measured data decreases the magnitude of relative deviations. To obtain a quantitative measure of the magnitude of the expected relation, non-linear quantile regression was performed for various quantiles of the dependencies using the two-parameter reciprocal function expressed in Eq 8 [25].

$$y = a + 1/(b \cdot x) \tag{8}$$

As the relative deviations in lateral and longitudinal position and in activity of the source could be positive or negative, quantiles describing a "SNR funnel" (5, 50 and 95%) were selected. In 90% of the cases, the estimates should fall on either side of the 50% quantile line,

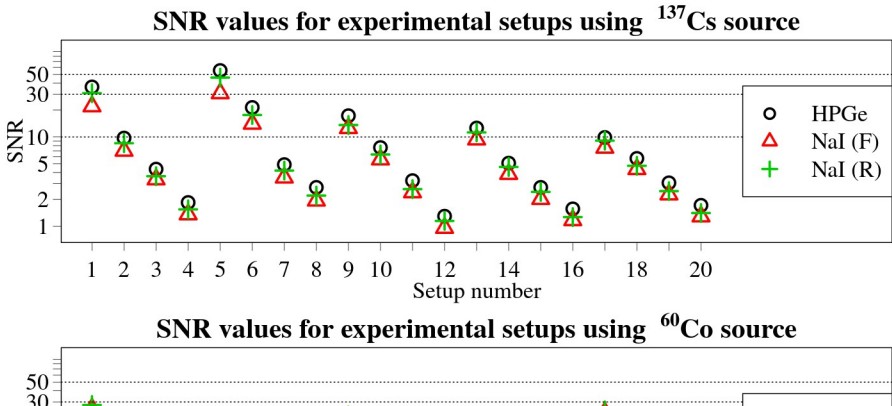

**Fig 9. Calculated SNRs for all experimental set-ups and detector combinations for the two radionuclides used.** Black circles mark values for HPGe detector, red triangles—front NaI(Tl) and green crosses—rear NaI(Tl) SNR values. Dotted horizontal lines mark SNR values of 10, 30 and 50. The SNR axis is logarithmic.

and between the bounds of these 5 and 95% quantile lines. Thus, for 5 and 95% quantile lines, the values *a* displays the horizontal asymptotes of the 90% confidence interval of the expected deviations, when the SNR of the data tends to infinity. The value *a* for the 50% quantile denotes a horizontal asymptote, to which tends the average deviation of an estimate, while the SNR of the data tends to infinity. These values a can visualise a trend to underestimate or over-estimate the parameter of interest, and at which distance are the limits of the 90% confidence interval from the median. Because the relative deviation in position is expressed as a distance between the estimated and actual position of the source, it can only obtain positive values. Thus, it is necessary to perform the regression only for the 90% quantile to cover 90% of the most probable estimates.

To evaluate the maximum absolute expected deviations for relative deviations of lateral position, longitudinal position and activity, distances between two sets of quantile lines were calculated. The first distance—between 50 and 5% quantile lines, the other—between 50 and 95% quantile lines. The larger of the two distances was selected as a value for a given SNR. For relative deviation of position—as a distance between the *x* axis and the 90% quantile line for a given SNR.

The Bayesian algorithm was running for 30 000 MCMC iterations using the experimental data. The first 10 000 iterations were discarded as burn-in.

## Results

### Assessment of the point estimate approximation of the detector count-rate function

Box plots of relative deviations in activity, lateral and longitudinal position of the source obtained using point and integrated estimates of count-rate function in the likelihood calcula-tions for experimental set-up 1, which consisted of a $^{137}$Cs source positioned 10 metres from the road and a $^{60}$Co source 20 m away, are shown in Fig 10.

As the change in the shape of the peak in the measurement time series is almost symmetric (only slight differences occur in the front and back of the peak due to the angular variations in counting efficiency of the detector in the vehicle), the deviations caused by using point

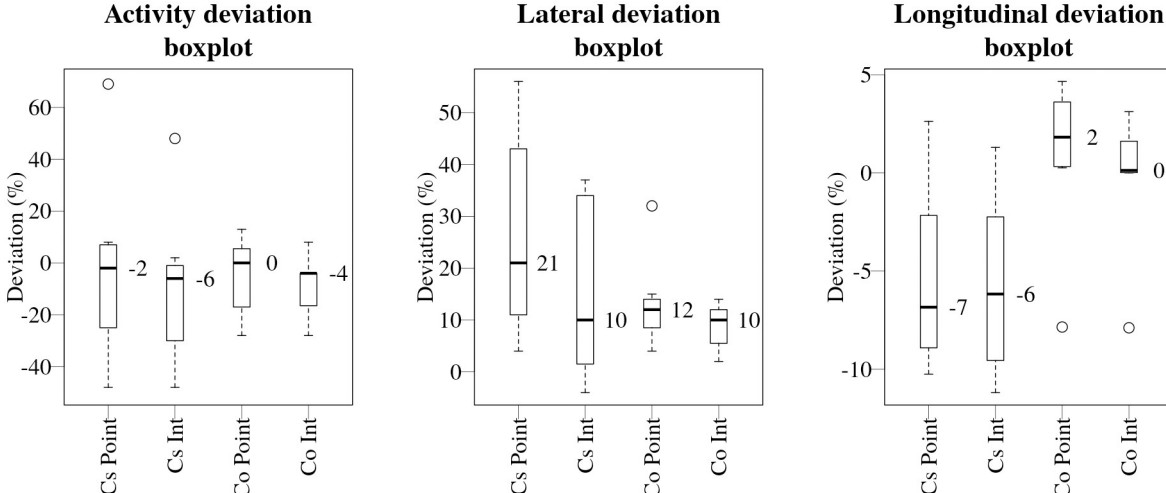

**Fig 10. Boxplot comparison for activity, lateral and longitudinal (from left to right) deviations obtained using point and integrated estimates for $^{137}$Cs and $^{60}$Co radionuclides.** Median values are marked with thick black lines and expressed in numbers to the right of the median line.

estimates of count-rate function should mainly affect the lateral position of the source. This was indeed the case. The median values of the lateral deviations for the $^{137}$Cs source using point estimates was 21%, which can be compared with 10% when using integrated estimates of the count-rate function in the likelihood calculations. For the $^{60}$Co source, on the other hand, the lateral deviations obtained using point estimates were 12% which was only slightly higher than the value obtained using integrated estimates (10%). This discrepancy is expected, as the $^{137}$Cs source was positioned 10 metres from the road, near the limit where the point-by-point deviations start to increase rapidly (15% at 10 metres), while the $^{60}$Co source was positioned 20 metres away, where the point-by-point deviations are about three times smaller (no more than 5% in the worst case). The differences between the median values of activity and longitudinal deviations for $^{137}$Cs and $^{60}$Co are within a few percent of each other when comparing the results obtained using point and integrated estimates of the count-rate function. The greatest difference was of 2% between all of the activity and longitudinal deviation comparisons.

The fact that only relative deviations of lateral position of the $^{137}$Cs source positioned at 10 m distance were affected when comparing the use of point and integrated estimates confirms that it is indeed possible to use the point approximation of the count-rate function in the likelihood calculations and to obtain results of comparable accuracy using less computational resources, providing the distance travelled by the source during the acquisition interval is smaller than the distance to the source.

## Effects of the discrepancy in the coordinates of the measurement position on the Bayesian estimates

Due to the fact that the discrepancies in the measurement coordinates could potentially affect the estimated position, and even the activity of the source, two Bayesian estimates were compared using modelled data: one with the exact coordinates of the measurements, and the other with the reverse-engineered coordinate shifts. The combined posterior distributions, of position and activity of the source, throughout the different chains displays an approximate average effect, which was caused by the synthetic deviations of measurement coordinates. The resulting combined posterior distributions for source position and activity are shown in Figs 11 and 12 respectively.

It can be seen from Fig 11 that the width of posterior distribution of the longitudinal coordinate increased by more than three times—the distances between the 2.5% and 97.5% quantiles of the longitudinal coordinate distributions were 179% and 52% for using the synthetic deviations and not using the deviations of the coordinates correspondingly. The shape of the posterior distribution of longitudinal position of the source suggests that the discrepancies introduced in the measurement coordinates can result in a shift of the most probable point of the source along the road, providing a precise, but inaccurate, estimate of the longitudinal position of the source. It is possible to observe a slight shift in the activity value of the highest probability in the posterior distribution—a reduction of the estimated activity value by 7.1%. The shape of the posterior distributions for the lateral position has not changed.

The comparison using modelled data suggests, that such deviations in the measurement coordinates can potentially affect the localization of the source along the road, and even reduce the estimated activity by about 7%.

It must be noted, that the distribution of probabilities between the two sides of the road within the posterior probability distributions of source position (as seen in Fig 11) does not represent the average outcome of the Bayesian evaluation. In fact, the distribution of relative probability can vary depending on a particular data set, as can be seen in Fig 13, where posterior distributions for $^{60}$Co source position from experimental setup 5 are displayed for four

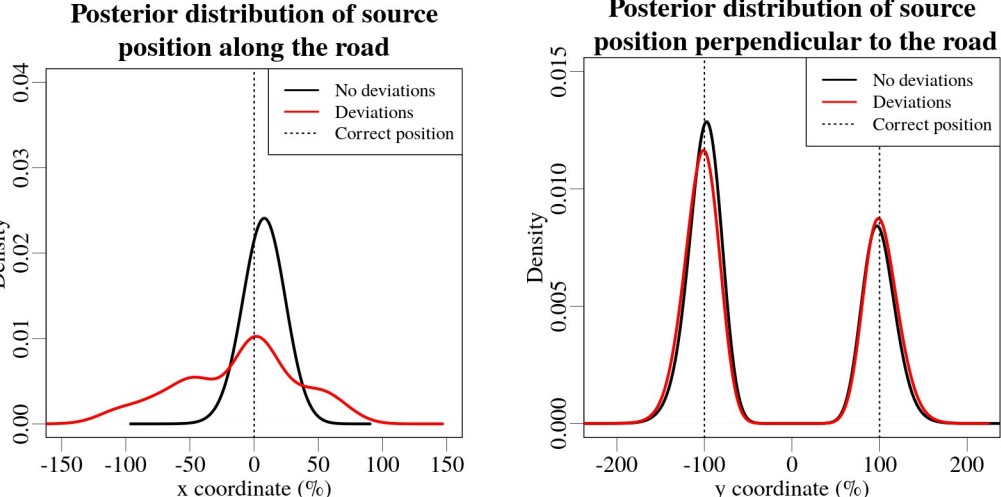

**Fig 11. Posterior distributions for longitudinal (left) and lateral (right) coordinates for modelled data without (black line) and with (red line) the synthetic deviations in the measurement coordinates.** The vertical dashed lines marks the correct positions of the source. The coordinates are expressed in terms of percent of the distance to the source (30 m).

different passes of the radioactive source. The distribution of relative probability between the two sides of the road in posterior distributions for source position can be seen varying from concentrating only on one side of the road to anything in between. The particular distribution of relative probabilities is a result of the interaction of a particular data set and the Bayesian model.

If the data and the model does not provide with enough information for the algorithm to choose a particular side of the road, then the posterior for the position of the source should have fairly equal amounts of relative probability on either side of the road. On the other hand,

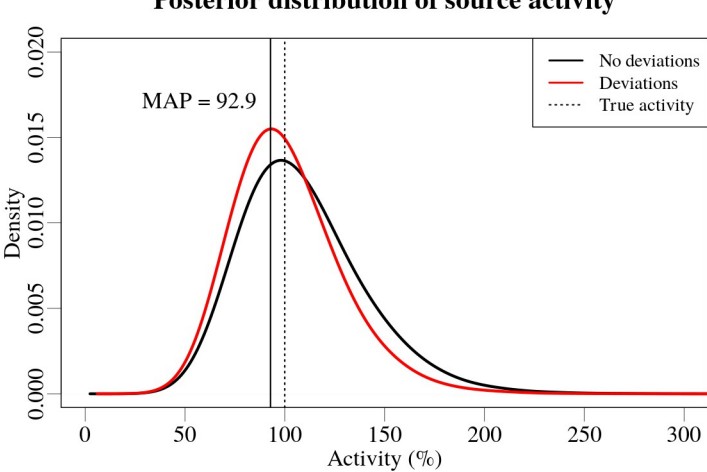

**Fig 12. Posterior distributions for activity of the source for simulated data without (black) and with (red) the synthetic deviations in the measurement coordinates.** The vertical dashed line marks the true activity of the source. The activity values are expressed in terms of percent of the activity of the source (100 MBq).

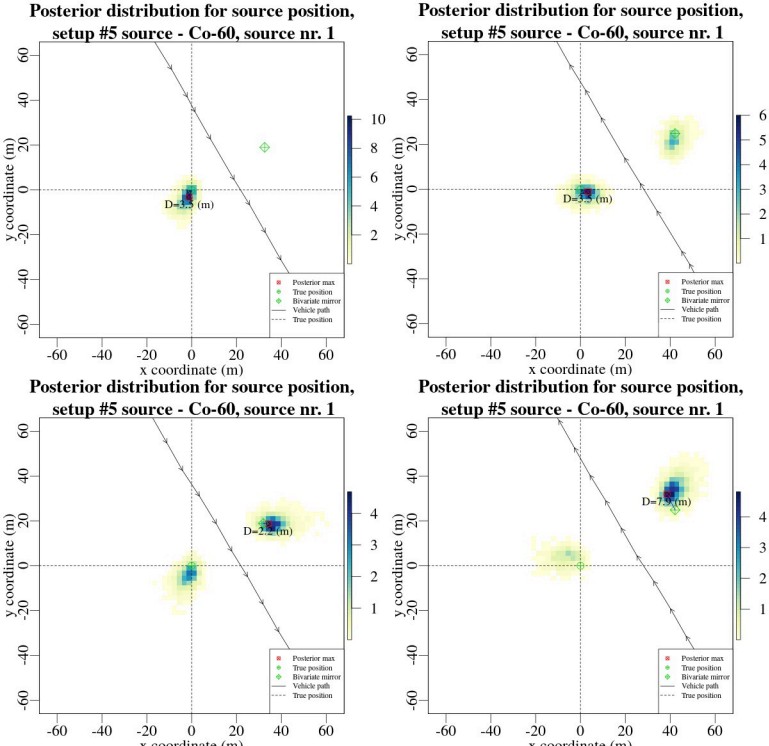

**Fig 13. Posterior probability distributions for source position for experimental setup 5, $^{60}$Co source, obtained from data collected using only HPGe detector while passing the source in both directions.** The colour within the posterior distributions denotes the relative probability of the source being there—the darker the colour the higher the probability. The crossed dashed lines marks the actual position (True position, as marked in the legend) of the radiation source. The arrows marks the geographical coordinates recorded using GNSS system each second.

if there was additional information within the Bayesian model or the data itself that allows to deconstruct more information regarding the position of the source (for example the angular calibration of the counting efficiency of the detectors that was included within the Bayesian model), then depending on the amount of additional information the posterior distribution might have more or less relative probability for each side of the road, leading to asymmetric posterior distributions.

For this to be true, there must be equal possibilities for the algorithm to sample both sides of the road. Otherwise, if the initial samples within the MCMC chain were leading towards one side of the road, the chain might get stuck on that particular side of the road. This might happen irrespective of the particular data set or the chosen Bayesian model. To counter this possible "bias" of the algorithm, the discussed method of moving the MCMC proposal to the other side of the road each 1000 MCMC iterations was implemented, as described in [13] in more detail.

It could be argued, that some kind of data pre-processing before feeding it into the Bayesian algorithm would improve the results—for example replacing the errorneous coordinates by interpolating the measurement coordinates without the discrepancies. Depending on the manifestation of the discrepancies, this could potentially increase the accuracy and precision of the Bayesian estimates. Tests using this experimental data did not yield significant improvements in accuracy and precision of the Bayesian estimates.

Another possible solution could employ usage of, for example, inertial navigation systems, which would provide more accurate information regarding the positioning of the vehicle. Further investigations and experiments would be required to confirm the gains and practical aspects of using such system, as it can not be said for certain that better systems will have no flaws.

Despite the chosen methodology to reduce the discrepancies, such improved systems can potentially be significantly more complicated compared to a simple system using a single GNSS receiver, which could in turn limit the real-time applicability of the method. Additionally, manifestations of such discrepancies in other mobile gamma spectrometry systems might be different, possibly requiring different approaches to this problem.

## Evaluation of the influence of the SNR of the measurement data to the deviations in estimates

The relative deviations in activity and position for all the experimental set-ups and source and detector combinations are plotted against their respective total linearly combined SNRs for the various combinations of sources and detectors in Figs 14 and 15 respectively. Correspondingly, the relative deviations in lateral and longitudinal position are shown in Figs 16 and 17.

The inverse relationship between the magnitude of the relative deviations and the SNR is indeed visible to varying degrees, as displayed by the $b$ values obtained during quantile regression. This is visualized in Figs 14–17 by black lines representing the selected quantiles. The larger the $b$ value, the less steep the quantile line, the smaller the SNR effect on the deviation of the result. Thus, the trend is strongest for the relative deviations in activity ($b_{5\%}$ = 0.006, $b_{95\%}$ = 0.003). For the position ($b_{90\%}$ = 0.013) and lateral deviations ($b_{5\%}$ = 0.02, $b_{95\%}$ = 0.011) the trend was less evident. The weakest trend was observed for the relative deviations in longitudinal position, having the highest values ($b_{5\%}$ = 0.069, $b_{95\%}$ = 0.058).

The $a$ values, describing the asymptote of the relationship between the 50% quantile lines and SNR of the data, indicates that, on average, the estimates of the lateral and longitudinal positions do not deviate significantly from the actual position of the source ($a_{lat}$ = 0.6; $a_{long}$ =

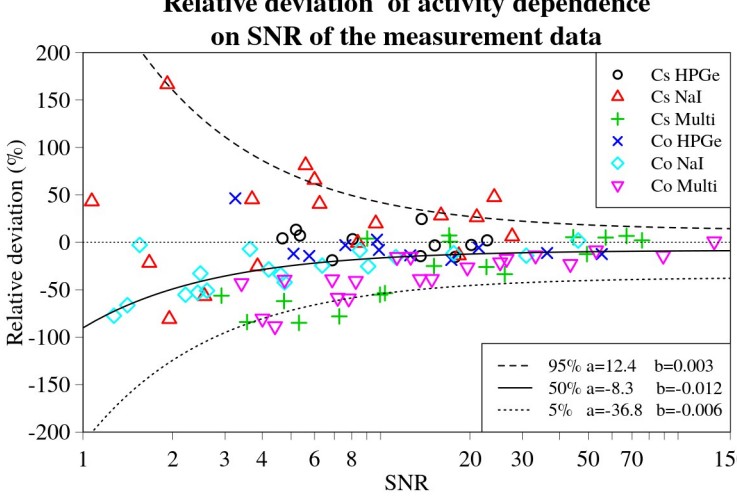

**Fig 14. Activity relative deviation values for every experimental set-up, detector and source combination plotted against the respective SNR values.** Black dashed line denotes the 5%, solid line—50% and dotted line—95% (from top to bottom) curves denoting corresponding quantile dependencies on SNR of the data, obtained using non-linear quantile regression. Fitted parameters $a$ and $b$ of the lines are displayed in a legend on the right.

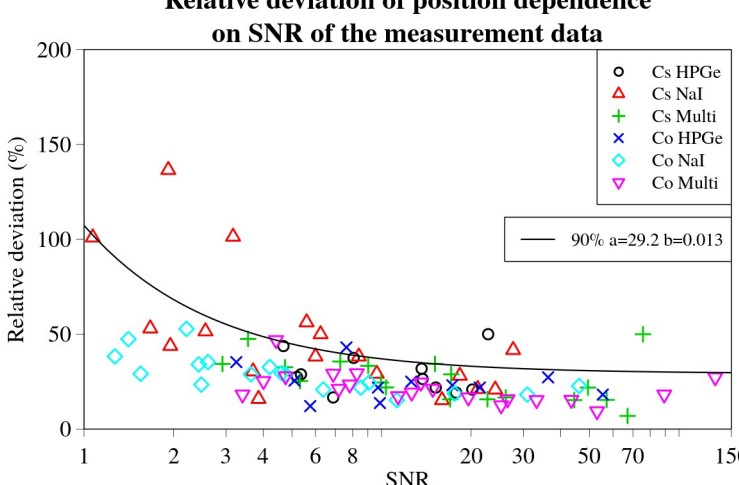

**Fig 15. Position relative deviation values for every experimental set-up, detector and source combination plotted against the respective SNR values.** Black line denotes a curve representing the 90% quantile dependency on the SNR of the data obtained using non-linear quantile regression. Fitted parameters *a* and *b* of the line are displayed in a legend on the right.

0.3), as illustrated in Figs 16 and 17. On the other hand, the *a* value for the deviations in activity $a_{act} = -8.3$, indicates a slight underestimation of the activity by, on average, about 8%. This small, constant deviation in the estimated activity could be due to a number of different factors affecting the experimental data. It may also depend on the offsets detected in the GNSS positioning system readout, as shown in Fig 12.

An outlier can clearly be seen in the data ($x_{SNR} = 23$; $y_{dev} = -42\%$) for longitudinal deviations of $^{137}$Cs source using a HPGe detector in Fig 17, being positioned significantly outside the main "funnel" of the 5 and 95% quantiles. This point belongs to the experimental set-up 1

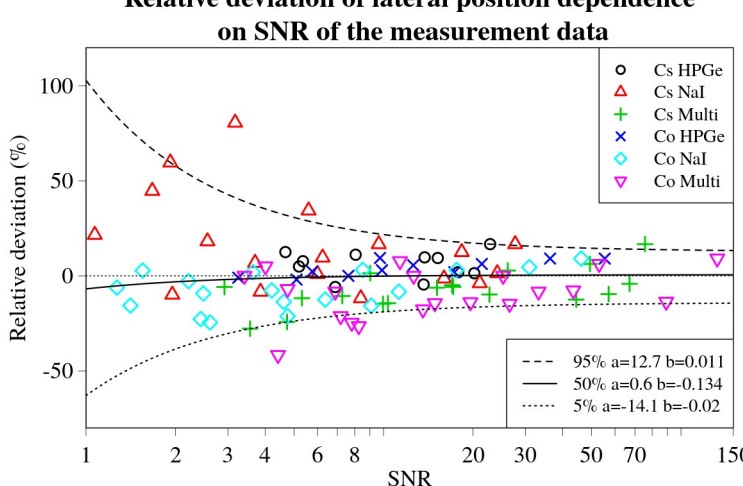

**Fig 16. Lateral position relative deviation values for every experimental set-up, detector and source combination plotted against the respective SNR values.** Black dashed line denotes the 5%, solid line—50% and dotted line—95% (from top to bottom) curves denoting corresponding quantile dependencies on SNR of the data, obtained using non-linear quantile regression. Fitted parameters *a* and *b* of the lines are displayed in a legend on the right.

**Relative deviation of longitudinal position dependence on SNR of the measurement data**

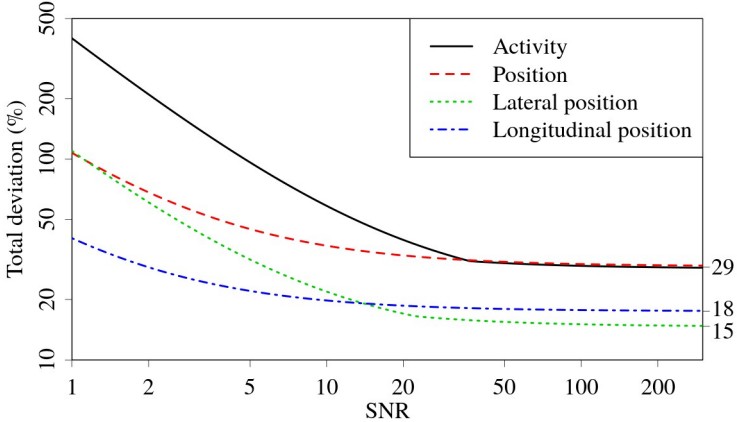

**Fig 17. Longitudinal position relative deviation values for every experimental set-up, detector and source combination plotted against the respective SNR values.** Black dashed line denotes the 5%, solid line—50% and dotted line—95% (from top to bottom) curves denoting corresponding quantile dependencies on SNR of the data, obtained using non-linear quantile regression. Fitted parameters *a* and *b* of the lines are displayed in a legend on the right.

(distance to the source 10 m), and because the graph is displayed in percentage of the distance to the source, relatively small deviations (7 m) can obtain high values of relative deviations.

The maximum absolute expected deviation from the reference line starts to approach the horizontal asymptote at around SNR 20-30 for all the quantile curves, as shown in Fig 18. It is possible to observe knees in the curves for lateral position and activity deviations in the vicinity of SNR region (20-30). If the distances between the reference line and the 5 and 95% quantile lines are different, there might be a point along the SNR axis where the distance to a particular quantile line becomes larger than the distance to the other quantile line. At this point, a knee

**Dependency of the maximum absolute expected deviation from the reference line on SNR of the measurement data**

**Fig 18. Graphs of maximum absolute expected deviation from the reference line (between 5%, 95% and 50% quantiles for activity—solid black, lateral position—dotted green, longitudinal position—dot-dashed blue, and between the *x* axis and the 90% quantile for combined position deviations—dashed red).** Asymptote values for the curves are displayed on the right axis of the graph.

### Comparison of theoretical and experimental absolute relative deviations

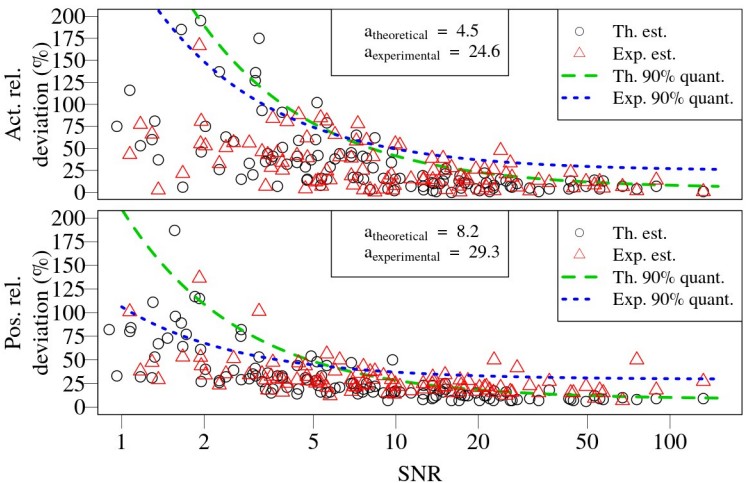

**Fig 19. Comparison of the asymptotic values of quantile regression for activity and position relative deviations obtained using theoretical data (from [13]) and experimental data.** The asymptotic values *a* are displayed in the legend in the top. The quantile regression line for theoretical data is marked with green dashed curve, while for the experimental data—blue dotted curve.

in the curve might be visible due to the differences in *a* and *b* parameters of the quantile curves. Despite this, it is still clearly visible that a further increase in the SNR of the data does not lead to a significant reduction in the expected maximum deviations. Thus, above values of 20-30, the SNR of the data is not the main factor limiting the performance of the Bayesian algorithm.

Based on the asymptote values of the total deviation curves (given at the right side of Fig 18), it is not unreasonable to assume that roughly 50% of the total relative deviation in source position (asymptote value of 29.2%) is due to deviations in the lateral position (asymptote value of 14.7%). The other 50% must therefore be due to deviations in the longitudinal measurement coordinates.

Comparison of the absolute relative deviations for activity and position of the source obtained using experimental data with relative deviations obtained using theoretical data (from [13]) displayed in Fig 19. It can be seen, that in 90% of the cases, the relative deviations for activity and position of the source using the theoretical data will be smaller than 4.5 and 8.2%, while for experimental data—24.6 and 29.3% respectively. This increase of 3-5 times in the expected absolute relative deviations in 90% of the cases is anticipated due to additional uncertainties pertaining the experimental data compared to the simulated, theoretical data from [13] (e.g. discrepancies in measurement coordinates, varying background radiation, gain drift in the detectors).

All of the measurements used throughout this study were performed at a constant speed of 50 km/h. In real-world situations this might not necesseraly be the case. Theoretically, the higher the speed of the detector past the radioactive source, the less time spent in the vicinity of the source, the lower the quality and quantity of the information collected (lower number of counts) thus the more uncertain the Bayesian estimates. To minimize the uncertainty for a given detector system, thus more time has to be spent in the vicinity of the source—either by performing a slower pass or multiple faster passes.

Despite the fact that known sources had been used throughout the studies and that their actual positions and activities were used in evaluating the results, no knowledge about the sources is

required to use this algorithm. It might be argued that the radioactive source could then be some other source than [137]Cs or [60]Co. In that case advances in computational resources could be utilized and simultaneous MCMC chains could be ran for a list of different radionuclides.

## Conclusions

The Bayesian inference method used in mobile gamma spectrometry for finding orphan sources has been shown to have the ability to determine source activity and location of unshielded sources using experimental data. The method was tested using point sources of [60]Co and [137]Cs at distances between 10 and 190 m from the road. A 123% HPGe spectrometer and two 4-litre NaI(Tl) spectrometers were used in a measuring vehicle, driving at a speed of 50 km/h, using 1 s acquisition time intervals. Source activities and locations were determined using three measuring data sets: data from the HPGe detector only, data from one NaI(Tl) detector and data from all three detectors combined.

The precision to determine activity and location of the sources was approximately the same for the HPGe and NaI(Tl) detectors. The use of the measurement data from the three detectors in combination slightly increased the accuracy. For well-detected sources with signal-to-noise ratios (SNR) exceeding 20, activity and location deviations from actual values were around 30%. A higher SNR did not significantly improve precision. At SNR values close to the detection limit (SNR less than 3), the deviation increased to 50–100% or more. About half of the uncertainty in the location prediction was due to deviation in the lateral coordinate (distance perpendicular to the road) and half due to uncertainty in the longitudinal coordinate (distance along the road).

If the navigation system does not provide coordinates with a precision greater than the length of a sampling interval (here 14 m), the uncertainty in the predictions will increase. Source photon recordings at the detector are added during the acquisition time interval as the measuring vehicle moves along the road. Measurement data are typically output at the end of the interval. This causes the spatial resolution to deteriorate. However, if the distance to the source is greater than the distance of one sampling interval, it has no significant effect on the precision of the activity and location determination of the source if point approximation of count-rate function is used in the likelihood calculations.

Because the Bayesian algorithm is implemented in **R** language, it is quite slow (3-5 minutes per estimation). Currently, there are no possibilities to read the data from the detectors in real time, thus the data has to be imported after the experimental survey. Additional limitation of the developed algorithm is that it has been tested in localisation of only one source at the moment in its current state.

Although the Bayesian approach used in this investigation was designed to locate a single source in two dimensions, the approach could probably be extended to locate multiple 3D sources as long as there is sufficient spatial resolution in the measuring positions. It may even be developed to map uneven deposition of radionuclides in radiological emergency situations.

In summary, our investigations using experimental data from a representative set of carborne detector systems confirms our previous predictions that Bayesian methods to project single source location and activity are highly useful for orphan source search in mobile gamma spectrometry.

## Acknowledgments

The authors thank Mats Hansson, Mattias Jönsson and Kurt Sundin for assisting with the measurements and Barsebäck Kraft AB, Barsebäck Gods AB and Löddeköpinge Rescue Services for access to areas for carrying out parts of the measurements.

## Author Contributions

**Conceptualization:** Antanas Bukartas, Jonas Wallin, Robert Finck, Christopher Rääf.

**Data curation:** Antanas Bukartas, Jonas Wallin, Robert Finck, Christopher Rääf.

**Formal analysis:** Antanas Bukartas, Jonas Wallin, Robert Finck, Christopher Rääf.

**Funding acquisition:** Christopher Rääf.

**Investigation:** Antanas Bukartas, Jonas Wallin, Robert Finck, Christopher Rääf.

**Methodology:** Antanas Bukartas, Jonas Wallin, Robert Finck, Christopher Rääf.

**Project administration:** Christopher Rääf.

**Resources:** Antanas Bukartas, Christopher Rääf.

**Software:** Antanas Bukartas, Jonas Wallin, Christopher Rääf.

**Supervision:** Jonas Wallin, Robert Finck, Christopher Rääf.

**Validation:** Antanas Bukartas, Jonas Wallin, Robert Finck, Christopher Rääf.

**Visualization:** Antanas Bukartas, Robert Finck, Christopher Rääf.

**Writing – original draft:** Antanas Bukartas.

**Writing – review & editing:** Antanas Bukartas, Jonas Wallin, Robert Finck, Christopher Rääf.

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
