## [Decision Letter · Decision Letter 0]

7 Jul 2021

PONE-D-21-18279

Accuracy of a Bayesian technique to estimate position and activity of orphan gamma-ray sources by mobile gamma spectrometry: Influence of imprecisions in positioning systems and computational approximations

PLOS ONE

Dear Dr. Bukartas,

Thank you for submitting your manuscript to PLOS ONE. After careful consideration, we feel that it has merit but does not fully meet PLOS ONE’s publication criteria as it currently stands. Therefore, we invite you to submit a revised version of the manuscript that addresses the points raised during the review process.

We look forward to receiving your revised manuscript.

Kind regards,

Mohammadreza Hadizadeh

Academic Editor

PLOS ONE

Reviewers' comments:

Reviewer's Responses to Questions

**Comments to the Author**

1. Is the manuscript technically sound, and do the data support the conclusions?

Reviewer #1: Partly

Reviewer #2: Partly

2. Has the statistical analysis been performed appropriately and rigorously? 

Reviewer #1: Yes

Reviewer #2: Yes

3. Have the authors made all data underlying the findings in their manuscript fully available?

Reviewer #1: Yes

Reviewer #2: No

4. Is the manuscript presented in an intelligible fashion and written in standard English?

Reviewer #1: Yes

Reviewer #2: Yes

5. Review Comments to the Author

Reviewer #1: The present paper is the continuation of the author’s previous work that was published in PLOS ONE. In this paper, authors investigated the accuracy of their proposed technique for estimating the activity and position of the source that has been lost. The present work is interesting and also important as it involves actual experiment to test the proposed theory. However, there are few issues that needs to be addressed. Please see my comments as attached and make changes to the current version of the manuscript. Considering my comments, I recommend minor revision of the manuscript before being considered for publication in PLOS ONE.

Reviewer #2: The article is thorough and well written. At this point there are some major and minor revisions needed to make it more complete and clear, which I describe in the attachment.

The data availability is listed as "The data will be prepared and made available during further submission process of the article," so I have checked "no" since I have not yet seen it.

6. PLOS authors have the option to publish the peer review history of their article (what does this mean?). If published, this will include your full peer review and any attached files.

Reviewer #1: No

Reviewer #2: No

---

## [Author Response · Author response to Decision Letter 0]

11 Oct 2021

The authors thank the reviewers for the time and help in improving the manuscript.

All of the reviewers comments are replied within the supplied file with the name "Answer_to_comments_2021-09-03.pdf".

---

## [Decision Letter · Decision Letter 1]

1 Nov 2021

PONE-D-21-18279R1Accuracy of a Bayesian technique to estimate position and activity of orphan gamma-ray sources by mobile gamma spectrometry: Influence of imprecisions in positioning systems and computational approximations

Dear Dr. Bukartas,

Thank you for submitting your manuscript to PLOS ONE. Reviewers' comments on your work have now been received. Comments from the reports appear below. These comments suggest that specific revisions of your manuscript are in order. When you resubmit your manuscript, please include a summary of the changes made and a succinct response to all recommendations or criticisms contained in the reports.

We look forward to receiving your revised manuscript.

Kind regards,

Mohammadreza Hadizadeh

Academic Editor

PLOS ONE

Reviewers' comments:

Reviewer's Responses to Questions

**Comments to the Author**

1. If the authors have adequately addressed your comments raised in a previous round of review and you feel that this manuscript is now acceptable for publication, you may indicate that here to bypass the “Comments to the Author” section, enter your conflict of interest statement in the “Confidential to Editor” section, and submit your "Accept" recommendation.

Reviewer #1: All comments have been addressed

Reviewer #2: (No Response)

2. Is the manuscript technically sound, and do the data support the conclusions?

Reviewer #1: Yes

Reviewer #2: Partly

3. Has the statistical analysis been performed appropriately and rigorously? 

Reviewer #1: Yes

Reviewer #2: Yes

4. Have the authors made all data underlying the findings in their manuscript fully available?

Reviewer #1: Yes

Reviewer #2: No

5. Is the manuscript presented in an intelligible fashion and written in standard English?

Reviewer #1: Yes

Reviewer #2: Yes

6. Review Comments to the Author

Reviewer #1: The authors addressed all my comments very well. The revised manuscript is now okay. Therefore, I recommend the present paper to be accepted for publication in PLOS ONE.

Reviewer #2: Please see the attached comments. I feel that the presence of the position errors from the software issue are still a weakness of the analysis, and that the data to be considered "fully available" need metadata and possibly a more universal format.

7. PLOS authors have the option to publish the peer review history of their article (what does this mean?). If published, this will include your full peer review and any attached files.

Reviewer #1: No

Reviewer #2: No

---

## [Author Response · Author response to Decision Letter 1]

16 Feb 2022

All of the comments are answered in an attached file.

---

## [Decision Letter · Decision Letter 2]

8 Mar 2022

PONE-D-21-18279R2Accuracy of a Bayesian technique to estimate position and activity of orphan gamma-ray sources by mobile gamma spectrometry: Influence of imprecisions in positioning systems and computational approximationsPLOS ONE

Dear Dr. Bukartas,

Reviewers' comments on your revised work have now been received. Comments from the reports appear below. One of the referees suggests specific minor revisions of your manuscript. When you resubmit your manuscript, please include a summary of the changes made and a succinct response to all recommendations or criticisms contained in the reports.

We look forward to receiving your revised manuscript.

Kind regards,

Mohammadreza Hadizadeh

Academic Editor

PLOS ONE

Journal Requirements:

Reviewers' comments:

Reviewer's Responses to Questions

**Comments to the Author**

1. If the authors have adequately addressed your comments raised in a previous round of review and you feel that this manuscript is now acceptable for publication, you may indicate that here to bypass the “Comments to the Author” section, enter your conflict of interest statement in the “Confidential to Editor” section, and submit your "Accept" recommendation.

Reviewer #1: All comments have been addressed

Reviewer #2: (No Response)

2. Is the manuscript technically sound, and do the data support the conclusions?

Reviewer #1: Yes

Reviewer #2: Partly

3. Has the statistical analysis been performed appropriately and rigorously? 

Reviewer #1: Yes

Reviewer #2: Yes

4. Have the authors made all data underlying the findings in their manuscript fully available?

Reviewer #1: Yes

Reviewer #2: Yes

5. Is the manuscript presented in an intelligible fashion and written in standard English?

Reviewer #1: Yes

Reviewer #2: Yes

6. Review Comments to the Author

Reviewer #1: Authors addressed all the comments very well. The manuscript quality has been improved significantly. I agree with the authors that "list of different radionuclides could be used when the orphan source is something other than Cs-137 or Co-60". It would be really interesting if authors can develop a library for various radioactive sources in their future works so the present method can be used to accurately locate different orphan sources.

I have no other comments on this paper, therefore I recommend the paper to be accepted for publication in PLOS ONE.

Reviewer #2: Thank you for your effort in addressing my comments. I have two remaining minor issues, continuing the numbering from before:

(1g) Thank you for reanalyzing some of the data using my suggestion to interpolate the data to eliminate the discrepancies, and for commenting on this in the paper. However, seeing those 2D plots of the position posterior gives rise to some more questions. From the paper, specifically the references to a bimodal posterior, also Figure 11, and the discussion of the choice to “mirror” the source locations across the road, I was under the impression that the posterior was always fairly symmetric across the road. Figure 11 especially gives the impression that not only is the posterior symmetric, but deviations in the data do not strongly affect the symmetry of the posterior across the road. But both of these plots show posterior distributions that are strongly asymmetric, with peak probabilities in a 2-to-1 ratio (assuming a linear scale) for the original analysis, and peak ratios of greater than 9-to-1 for the interpolated track. Can you please (A) be clear what the spatial posterior distributions look like in practice, by perhaps (B) including a figure showing the 2D posterior distribution, and (C) explain in the paper why the posteriors can be so asymmetric, and (D) better justify the decision to mirror the results across the track, in light of this asymmetry.

(14) Thank you for the clarification that the code to read the data is in the separate GitHub repository at https://github.com/SpontaneousFusion/ImpNBL. Please include a reference to this repository in the data repository (perhaps in the README.md file) since it is not easily located from the data repository, and the only metadata provided merely says “Within the archive you will find AUTOMORC experiment data.” You may also want to include a reference to your paper once it comes out.

7. PLOS authors have the option to publish the peer review history of their article (what does this mean?). If published, this will include your full peer review and any attached files.

Reviewer #1: No

Reviewer #2: No

---

## [Author Response · Author response to Decision Letter 2]

25 Apr 2022

The response to the reviewers questions can be found within the attached file.

---

## [Decision Letter · Decision Letter 3]

3 May 2022

Accuracy of a Bayesian technique to estimate position and activity of orphan gamma-ray sources by mobile gamma spectrometry: Influence of imprecisions in positioning systems and computational approximations

PONE-D-21-18279R3

Dear Dr. Bukartas,

We’re pleased to inform you that your manuscript has been judged scientifically suitable for publication and will be formally accepted for publication once it meets all outstanding technical requirements.

Kind regards,

Mohammadreza Hadizadeh

Academic Editor

PLOS ONE

Additional Editor Comments (optional):

Reviewers' comments:

Reviewer's Responses to Questions

**Comments to the Author**

1. If the authors have adequately addressed your comments raised in a previous round of review and you feel that this manuscript is now acceptable for publication, you may indicate that here to bypass the “Comments to the Author” section, enter your conflict of interest statement in the “Confidential to Editor” section, and submit your "Accept" recommendation.

Reviewer #1: All comments have been addressed

Reviewer #2: All comments have been addressed

2. Is the manuscript technically sound, and do the data support the conclusions?

Reviewer #1: Yes

Reviewer #2: Yes

3. Has the statistical analysis been performed appropriately and rigorously? 

Reviewer #1: Yes

Reviewer #2: Yes

4. Have the authors made all data underlying the findings in their manuscript fully available?

Reviewer #1: Yes

Reviewer #2: Yes

5. Is the manuscript presented in an intelligible fashion and written in standard English?

Reviewer #1: Yes

Reviewer #2: Yes

6. Review Comments to the Author

Reviewer #1: Authors addressed all comments very well. I have no other comments. Therefore, I recommend the paper to be accepted for publication.

Reviewer #2: (No Response)

7. PLOS authors have the option to publish the peer review history of their article (what does this mean?). If published, this will include your full peer review and any attached files.

Reviewer #1: No

Reviewer #2: No

---

## [Editor Report · Acceptance letter]

13 Jun 2022

PONE-D-21-18279R3 

Accuracy of a Bayesian technique to estimate position and activity of orphan gamma-ray sources by mobile gamma spectrometry: Influence of imprecisions in positioning systems and computational approximations 

Dear Dr. Bukartas:

I'm pleased to inform you that your manuscript has been deemed suitable for publication in PLOS ONE. Congratulations! Your manuscript is now with our production department. 

Kind regards, 

on behalf of

Dr. Mohammadreza Hadizadeh 

Academic Editor

PLOS ONE